# CMOS plus stochastic nanomagnets enabling heterogeneous computers for probabilistic inference and learning

Nihal Sanjay Singh [1,11], Keito Kobayashi[1,2,3,11], Qixuan Cao[1,11], Kemal Selcuk[1], Tianrui Hu[1], Shaila Niazi[1], Navid Anjum Aadit [1], Shun Kanai [2,3,4,5,6,7,8], Hideo Ohno [2,4,5,9], Shunsuke Fukami [2,3,4,5,9,10] ✉ & Kerem Y. Camsari [1] ✉

Extending Moore's law by augmenting complementary-metal-oxide semiconductor (CMOS) transistors with emerging nanotechnologies (X) has become increasingly important. One important class of problems involve sampling-based Monte Carlo algorithms used in probabilistic machine learning, optimization, and quantum simulation. Here, we combine stochastic magnetic tunnel junction (sMTJ)-based probabilistic bits (p-bits) with Field Programmable Gate Arrays (FPGA) to create an energy-efficient CMOS + X (X = sMTJ) prototype. This setup shows how asynchronously driven CMOS circuits controlled by sMTJs can perform probabilistic inference and learning by leveraging the algorithmic update-order-invariance of Gibbs sampling. We show how the stochasticity of sMTJs can augment low-quality random number generators (RNG). Detailed transistor-level comparisons reveal that sMTJ-based p-bits can replace up to 10,000 CMOS transistors while dissipating two orders of magnitude less energy. Integrated versions of our approach can advance probabilistic computing involving deep Boltzmann machines and other energy-based learning algorithms with extremely high throughput and energy efficiency.

With the slowing down of Moore's Law[1], there has been a growing interest in domain-specific hardware and architectures to address emerging computational challenges and energy efficiency, particularly borne out of machine learning and AI applications. One promising approach is the co-integration of traditional complementary metal-oxide semiconductor (CMOS) technology with emerging nano-technologies (X), resulting in CMOS + X architectures. The primary objective of this approach is to augment existing CMOS technology with novel functionalities, by enabling the development of physics-inspired hardware systems that realize energy-efficiency, massive parallelism, and asynchronous dynamics, and apply them to a wide range of problems across various domains.

Being named one of the top 10 algorithms of the 20th century[2], Monte Carlo methods have been one of the most effective approaches

[1]Department of Electrical and Computer Engineering, University of California Santa Barbara, Santa Barbara 93106 CA, USA. [2]Research Institute of Electrical Communication, Tohoku University, 2-1-1 Katahira, Aoba-ku, Sendai 980-8577, Japan. [3]Graduate School of Engineering, Tohoku University, 6-6 Aramaki Aza Aoba, Aoba-ku, Sendai 980-0845, Japan. [4]WPI Advanced Institute for Materials Research (WPI-AIMR), Tohoku University, 2-1-1 Katahira, Aoba-ku, Sendai 980-8577, Japan. [5]Center for Science and Innovation in Spintronics (CSIS), Tohoku University, 2-1-1 Katahira, Aoba-ku, Sendai 980-8577, Japan. [6]PRESTO, Japan Science and Technology Agency (JST), Kawaguchi 332-0012, Japan. [7]Division for the Establishment of Frontier Sciences of Organization for Advanced Studies at Tohoku University, Tohoku University, Sendai 980-8577, Japan. [8]National Institutes for Quantum Science and Technology, Takasaki 370-1207, Japan. [9]Center for Innovative Integrated Electronic Systems (CIES), Tohoku University, 468-1 Aramaki Aza Aoba, Aoba-ku, Sendai 980-0845, Japan. [10]Inamori Research Institute of Science (InaRIS), Kyoto 600-8411, Japan. [11]These authors contributed equally: Nihal Sanjay Singh, Keito Kobayashi, Qixuan Cao. ✉e-mail: s-fukami@riec.tohoku.ac.jp; camsari@ece.ucsb.edu

in computing to solve computationally hard problems in a wide range of applications, from probabilistic machine learning, optimization to quantum simulation. Probabilistic computing with p-bits[3] has emerged as a powerful platform for executing these Monte Carlo algorithms in massively parallel[4,5] and energy-efficient architectures. p-bits have been shown to be applicable to a large domain of computational problems, from combinatorial optimization to probabilistic machine learning and quantum simulation[6–8].

Several p-bit implementations that use the inherent stochasticity in different materials and devices have been proposed, based on diffusive memristors[9], resistive RAM[10], perovskite nickelates[11], ferroelectric transistors[12], single photon avalanche diodes[13], optical parametric oscillators[14] and others. Among alternatives sMTJs built out of low-barrier nanomagnets have demonstrated significant potential due to their ability to amplify noise, converting millivolts of fluctuations to hundreds of millivolts over resistive networks[15], unlike alternative approaches with amplifiers[16]. Another advantage of sMTJ-based p-bits is the continuous generation of truly random bitstreams without the need to be reset in synchronous pulse-based designs[17,18]. The possibility of designing energy-efficient p-bits using low-barrier nanomagnets has stimulated renewed interest in material and device research with several exciting demonstrations from nanosecond fluctuations[19–21] to a better theoretical understanding of nanomagnet physics[22–25] and novel magnetic tunnel junction designs[26,27].

Despite promising progress with hardware prototypes[28–32], large-scale probabilistic computing using stochastic nanodevices remains elusive. As we will establish in this paper, designing purely CMOS-based high-performance probabilistic computers suited to sampling and optimization problems is prohibitive beyond a certain scale (>1M p-bits) due to the large area and energy costs of pseudorandom number generators. As such, any large-scale integration of probabilistic computing will involve strong integration with CMOS technology in the form of CMOS+X architectures. Given the unavoidable device-to-device variability, the interplay between continuously fluctuating stochastic nanodevices (e.g., sMTJs) with deterministic CMOS circuits and the possible applications of such hybrid circuits remain unclear.

In this paper, we first introduce the notion of a heterogeneous CMOS + sMTJ system where the asynchronous dynamics of sMTJs control digital circuits in a standard CMOS field programmable gate array (FPGA). We view the FPGA as a "drop-in replacement" for eventual integrated circuits where sMTJs could be situated on top of CMOS. Unlike earlier implementations where sMTJs were primarily used to implement neurons and CMOS or analogue components circuits for synapses[28,29], we design hybrid circuits where sMTJ-based p-bits control a large number of digital circuits residing in the FPGA without dividing the system into neurons (sMTJ) and synapses (CMOS). We show how the true randomness injected into deterministic CMOS circuits augments low-quality random number generators based on linear feedback shift registers (LFSR). This result represents an example of how sMTJs could be used to reduce footprint and energy consumption in the CMOS underlayer. In this work, we present a small example of a CMOS + sMTJ system, however, similar systems can be scaled up to much bigger densities, leveraging the proven manufacturability of magnetic memory at gigabit densities. Our results will help lay the groundwork for larger implementations in the presence of unavoidable device-to-device variations. We also focus beyond the common use case of combinatorial optimization of similar physical computers[33], considering probabilistic inference and learning in deep energy-based models.

Specifically, we use our system to train 3-hidden 1-visible layer deep and unrestricted Boltzmann machines that entirely rely on the asynchronous dynamics of the stochastic MTJs. Second, we evaluate the quality of randomness directly at the application level through probabilistic inference and deep Boltzmann learning. This approach contrasts with the majority of related work, which typically conducts statistical tests at the single device level to evaluate the quality of randomness[21,34–38] (see Supplementary Notes VIII, XI, and XII for more randomness experiments). As an important new result, we find that the quality of randomness matters in machine learning tasks as opposed to optimization tasks that have been explored previously. Finally, we conduct a comprehensive benchmark using an experimentally calibrated 7-nm CMOS PDK and find that when the quality of randomness is accounted for, the sMTJ-based p-bits are about four orders of magnitude smaller in area and they dissipate two orders of magnitude less energy, compared to CMOS p-bits. We envision that large-scale CMOS + X p-computers (>>$10^5$) can be a reality in scaled-up versions of the CMOS + sMTJ type computers we discuss in this work.

### Constructing the heterogeneous p-computer
Figure 1 shows a broad overview of our sMTJ-FPGA setup along with device and circuit characterization of sMTJ p-bits. Unlike earlier p-bit demonstrations with sMTJs as standalone stochastic binary neurons, in this work, we use sMTJ-based p-bits to generate asynchronous and truly random clock sources to drive digital p-bits in the FPGA (Fig. 1a–c).

The conductance of the sMTJ depends on the relative angle $\theta$ between the free and the fixed layers, $G_{MTJ} \propto [1 + P^2 \cos(\theta)]$, where $P$ is the interfacial spin polarization. When the free layer is made out of a low barrier nanomagnet $\theta$ becomes a random variable in the presence of thermal noise, causing conductance fluctuations between the parallel (P) and the antiparallel (AP) states (Fig. 1d).

The five sMTJs used in the experiment are designed with a diameter of 50 nm and have a relaxation time of about 1–20 ms, with energy barriers of ≈14–17 $k_BT$, assuming an attempt time of 1 ns[39] (see Supplementary Note II). In order to convert these conductance fluctuations into voltages, we design a new p-bit circuit (Fig. 1b, e). This circuit creates a voltage comparison between two branches controlled by two transistors, fed to an operational amplifier. As we discuss in Supplementary Note III, the main difference of this circuit compared to the earlier 3 transistor/1MTJ design used in earlier demonstrations[28,29] is in its ability to provide a larger stochastic window to tune the p-bit (Fig. 1h) with more variation tolerance (see Supplementary Note IV).

Figure 1c, e, f, g show how the asynchronous clocks obtained from p-bits with 50/50 fluctuations are fed to the FPGA. Inside the FPGA, we design a digital probabilistic computer where a p-bit includes a lookup table (LUT) for the hyperbolic tangent function, a pseudorandom number generator (PRNG) and a digital comparator (see Supplementary Note V).

The crucial link between analog p-bits and the digital FPGA is established through the clock of the PRNG used in the FPGA, where a multitude of digital p-bits can be asynchronously driven by analog p-bits. As we discuss in Sections 3, 4, depending on the quality of the chosen PRNG, the injection of additional entropy through the clocks has a considerable impact on inference and learning tasks. The potential for enhancing low-quality PRNGs using compact and scalable nanotechnologies, such as sMTJs, which can be integrated as a BEOL (Back-End-Of-Line) process on top of the CMOS logic, holds significant promise for future CMOS + sMTJ architectures.

## Results
### Probabilistic inference with heterogeneous p-computers
In the p-bit formulation, we define probabilistic inference as generating samples from a specified distribution which is the Gibbs-Boltzmann distribution for a given network (see Supplementary Note I for details). This is a computationally hard problem[40], and is at the heart of many important applications involving Bayesian inference[41], training probabilistic models in machine learning[42], statistical physics[43] and many others[44]. Due to the broad applicability of probabilistic inference,

improving key figures of merit, such as probabilistic flips per second (sampling throughput) and energy-delay product for this task are extremely important.

To demonstrate this idea, we evaluate probabilistic inference on a probabilistic version of the full adder (FA)[45] as shown in Fig. 2a. The truth table of the FA is given in Fig. 2b. The FA performs 1-bit binary addition and it has three inputs (A, B, Carry in = $C_{in}$) and two outputs (Sum = S, and Carry out = $C_{out}$). The probabilistic FA can be described in a 5 p-bit, fully connected network (Fig. 2a). When the network samples from its equilibrium, it samples states corresponding to the truth table, according to the Boltzmann distribution.

We demonstrate probabilistic sampling on the probabilistic FA using the digital p-bits with standalone LFSRs (only using the FPGA), sMTJ-clocked LFSRs (using sMTJ-based p-bits and the FPGA), and standalone Xoshiro RNGs (only using the FPGA). Our main goal is to compare the quality of randomness obtained by inexpensive but low-quality PRNGs such as LFSRs[46] with sMTJ-augmented LFSRs and high-quality but expensive PRNGs such as Xoshiro[47] (see Supplementary Note VI).

Figure 2c shows the comparison of these three different solvers where we measure the Kullback-Leibler (KL) divergence[48] between the cumulative distribution based on the number of sweeps and the ideal Boltzmann distribution of the FA:

$$KL[P_{exp}||P_{ideal}] = \sum_x P_{exp}(x) \log \frac{P_{exp}(x)}{P_{ideal}(x)}, \quad (1)$$

where $P_{exp}$ is the probability obtained from the experiment (cumulatively measured), and $P_{ideal}$ is the probability obtained from the Boltzmann distribution. For LFSR (red line), the KL divergence saturates when the number of sweeps exceeds $N = 10^4$, while for sMTJ-clocked LFSR (blue line) and Xoshiro (green line), the KL divergence decreases with increasing the number of sweeps. The persistent bias of the LFSR is also visible in the partial histogram of probabilities measured at $N = 10^6$ sweeps as shown in Fig. 2d (see Supplementary Note VII for the full histograms). It is important to note here, that in our present context where sMTJs are limited to a handful of devices, we use sMTJ-based p-bits to drive low-quality

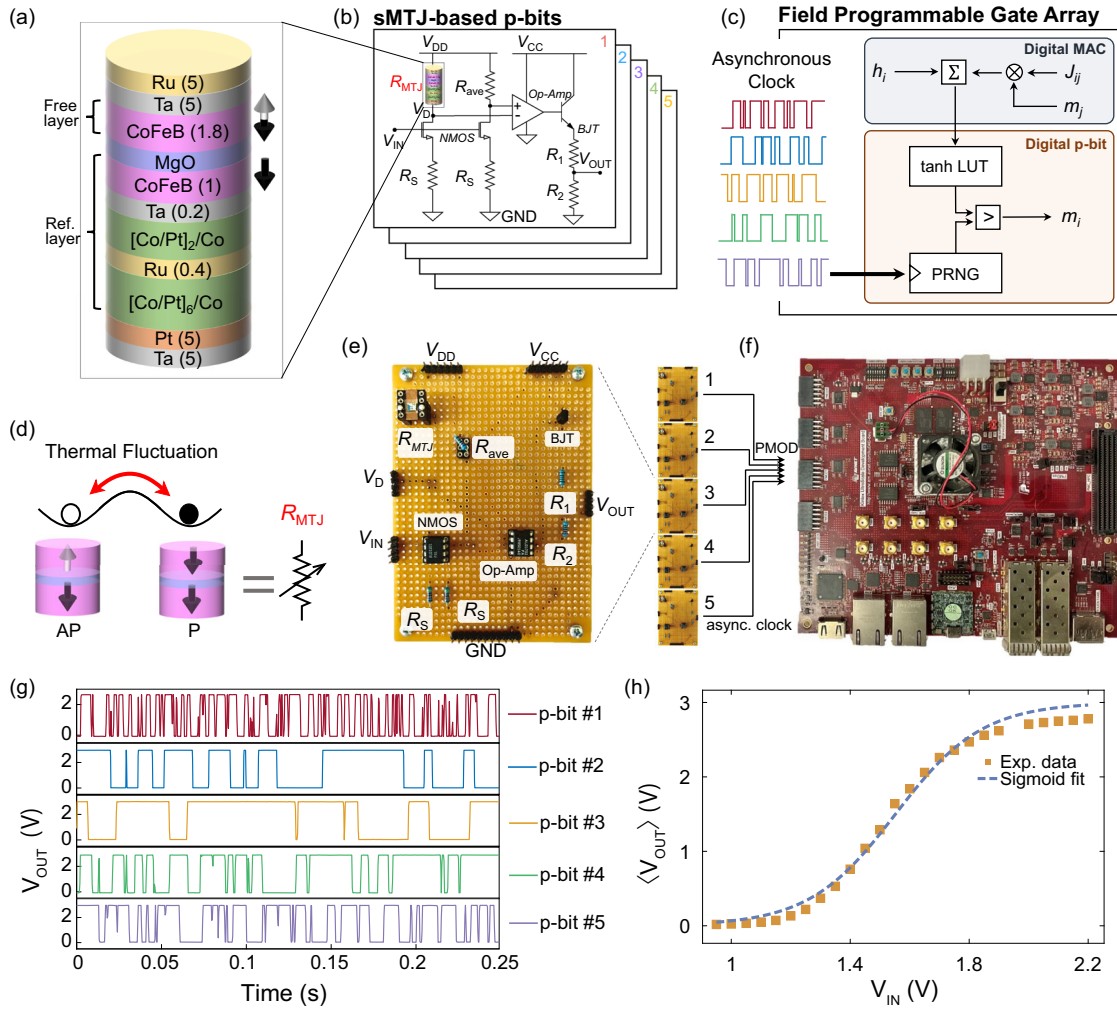

**Fig. 1 | Experimental setup for the CMOS + sMTJ probabilistic computer. a** Stack structure of the stochastic magnetic tunnel junction (sMTJ). **b** The proposed sMTJ-based p-bit circuit with two branches whose outputs are provided to an operational amplifier. $R_{ave}$ is the average resistance of $R_P$ and $R_{AP}$ of the sMTJ. 5 sMTJ-based p-bits provide tunable, truly random and asynchronous clocks to a digital field programmable gate array (FPGA). **c** Digital p-bits in the FPGA use lookup tables (LUT), comparators, synaptic weights, and pseudorandom number generators (PRNG). The clocks of the PRNG are driven by the truly random asynchronous outputs coming from the analog p-bits. **d** Pictorial representation of perpendicular sMTJ.

**e** Image of a single p-bit circuit. **f** Image of the FPGA. The asynchronous clocks are input through the peripheral module (PMOD) pins. **g** Typical output of p-bits #1 to 5 using 5 sMTJs obtained from the p-bit circuit (see Supplementary Note III), showing variations in fluctuations. **h** Experimentally measured $\langle V_{OUT} \rangle$ the time average (over a period of 3 minutes) of the p-bit circuit output, as a function of DC input voltage $V_{IN}$. The yellow squares are experimental data, and the blue dashed line is a fit of the form $\langle V_{OUT} \rangle = 1/2 \, V'_{CC}[\tanh[\beta(V_{IN} - V_0)] + 1]$, where $V_0 = 1.55$ V, $\beta = 3.43$ V$^{-1}$, $V'_{CC} = 3$ V is a reduced voltage from $V_{CC} = 5$ V (see Supplementary Note IV).

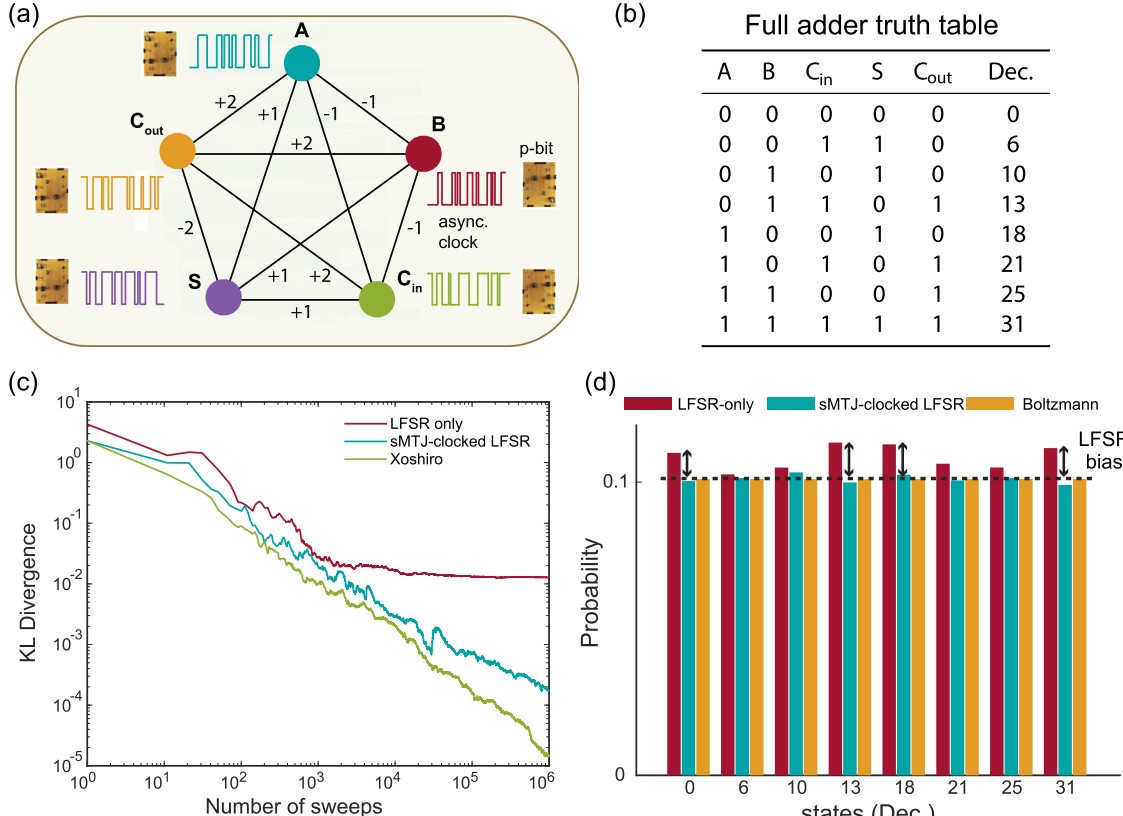

**(b) Full adder truth table**

| A | B | $C_{in}$ | S | $C_{out}$ | Dec. |
|---|---|---|---|---|---|
| 0 | 0 | 0 | 0 | 0 | 0 |
| 0 | 0 | 1 | 1 | 0 | 6 |
| 0 | 1 | 0 | 1 | 0 | 10 |
| 0 | 1 | 1 | 0 | 1 | 13 |
| 1 | 0 | 0 | 1 | 0 | 18 |
| 1 | 0 | 1 | 0 | 1 | 21 |
| 1 | 1 | 0 | 0 | 1 | 25 |
| 1 | 1 | 1 | 1 | 1 | 31 |

**Fig. 2 | Inference on a probabilistic full adder. a** Fully-connected full adder network[45], where p-bits are clocked by the sMTJs. **b** Truth table of the full adder where Dec. represents the decimal representation of the state of [A B $C_{in}$ S $C_{out}$] from left to right. **c** Kullback-Leibler (KL) divergence between the ideal and measured distributions vs. the number of sweeps. Results are shown for LFSR-based p-bit (red line), sMTJ-clocked LFSR-based p-bit (blue line), and Xoshiro-based p-bit

(green line). **d** Histogram for the measured and ideal distributions at the $10^6$ sweep. The red, blue, and yellow bars show LFSR, sMTJ-clocked LFSR, and Boltzmann distribution, respectively. The histogram shows all 8 high probability states denoted in (**b**) and with a clear bias for the LFSR distribution (see Supplementary Note VII for full histograms for all PRNGs, including Xoshiro).

LFSRs, observing how they perform similarly to high-quality PRNGs. In integrated implementations, however, sMTJ-based p-bits can be directly used as p-bits themselves without any supporting PRNG (see Supplementary Note XVI for details on projections of integrated implementations).

The mechanism of how the sMTJ-clocked LFSRs produce random numbers is interesting: even though the next bit in an LFSR is always perfectly determined, the randomness in the arrival times of clocks from the sMTJs makes their output unpredictable. Over the course of the full network's evolution, each LFSR produces an unpredictable bitstream, functioning as truly random bits.

The observed bias of the LFSR can be due to several reasons: first, the LFSRs generally provide low-quality random numbers and do not pass all the tests in the NIST statistical test suite[49] (see Supplementary Note XII). Second, we take whole words of random bits from the LFSR to generate large random integers. This is a known danger when using LFSRs[50,51], which can be mitigated by the use of phase shifters that scramble the parallelly obtained bits to reduce their correlation[52]. However, such measures increase the complexity of PRNG designs, further limiting the scalable implementation of digital p-computers (see Supplementary Note XI for detailed experimental analysis of LFSR bias).

The quality of randomness in Monte Carlo sampling is a rich and well-studied subject (see, for example, refs. 53–55). The main point we stress in this work is that even compact and inexpensive simple PRNGs can perform as well as sophisticated, high-quality RNGs when augmented by truly random nanodevices such as sMTJs.

## Boltzmann learning with heterogeneous p-computers

We now show how to train deep Boltzmann machines (DBM) with our heterogeneous CMOS + sMTJ computer. Unlike probabilistic inference, in this setting, the weights of the network are unknown, and the purpose of the training process is to obtain desired weights for a given truth table, such as the full adder (see Supplementary Note IX for an example of arbitrary distribution generation using the same learning algorithm). We consider this demonstration as a proof-of-concept for eventual larger-scale implementations (Fig. 3a, b). Similar to probabilistic inference, we compare the performance of three solvers: LFSR-based, Xoshiro-based and sMTJ+LFSR-based RNGs. We choose a 32-node Chimera lattice[56] to train a probabilistic full adder with 5 visible nodes and 27 hidden nodes in a 3-layer DBM (see Fig. 3b top panel). Note that this deep network is significantly harder to train than training fully visible networks whose data correlations are known a priori[29], necessitating positive and negative phase computations (see Supplementary Note VII and Algorithm 1 for details on the learning algorithm and implementation).

Figure 3c, d show the KL divergence and the probability distribution of the full adder Boltzmann machines based on the fully digital LFSR/Xoshiro and the heterogeneous sMTJ-clocked LFSR RNGs. The KL divergence in the learning experiment is performed like this: after each epoch during training, we save the weights in the classical computer and perform probabilistic inference to measure the KL distance between the learned and ideal distributions. The sMTJ-clocked LFSR and the Xoshiro-based Boltzmann machines produce probability distributions that eventually closely approximate the Boltzmann distribution of the full adder. On the other hand, the fully digital LFSR-

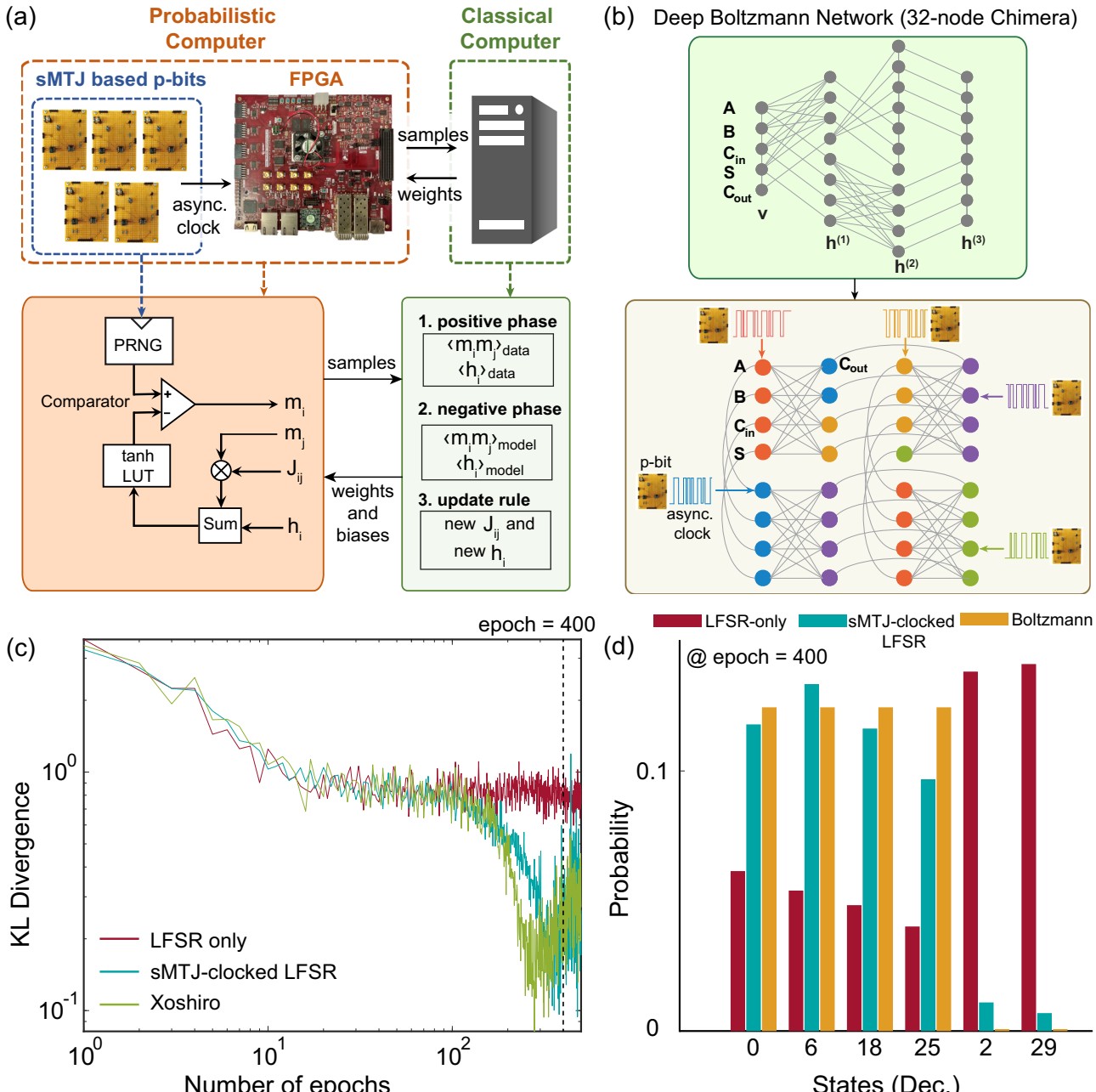

**Fig. 3 | Learning deep Boltzmann machines. a** The architecture of the p-computer for learning. The digital p-bits in FPGA are fed by sMTJ-based p-bits output similar to probabilistic inference. The weights $J_{ij}$ and biases $h_i$ are updated in the CPU for a specified number of epochs. **b** (Top) The 32-node Chimera graph is used as a deep BM. (Bottom) An asynchronous clocking scheme is shown with node coloring. **c** KL divergence as a function of the number of epochs for LFSR (red line), LFSR clocked by sMTJ-based p-bit (blue line), and Xoshiro (green line). **d** The distribution of full adder with learned weights and biases at epoch = 400 where the number of sweeps per epoch = 400 for LFSR-only and the number of sweeps per epoch = 16000 for sMTJ-clocked LFSR. The Boltzmann distribution was obtained with $\beta = 3$. The red, blue, and yellow bars show LFSR and LFSR clocked by sMTJ-based p-bit, and Boltzmann, respectively. The histogram shows 4 correct (0, 6, 18, 25) and 2 incorrect (2, 29) states, out of the 32 possible states. sMTJ-based p-bit closely approximates the ideal Boltzmann distribution, whereas the LFSR underestimates correct states and completely fails with states 2 and 29 (see Supplementary Note VII for full histograms for all PRNGs, including Xoshiro).

based Boltzmann machine produces the incorrect states [A B $C_{in}$ S $C_{out}$] = 2 and 29 with a significantly higher probability than the correct peaks, and grossly underestimates the probabilities of states 0, 6, 18, and 25 (see Supplementary Fig. 4 for full histograms that are avoided here for clarity). As in the inference experiment (Fig. 2a), the KL divergence of the LFSR saturates and never improves beyond a point. The increase in the KL divergence for Xoshiro and sMTJ-clocked LFSR towards the end is related to hyperparameter selection and unrelated to RNG quality[57]. For this reason, we select the weights at epoch=400 for testing to produce the histogram in Fig. 3d.

In line with our previous results, the learning experiments confirm the inferior quality of LFSR-based PRNGs, particularly for learning tasks (see Supplementary Note X for MNIST training comparisons between p-bits based on Xoshiro and LFSR). While LFSRs can produce correct peaks with some bias in optimization problems, they fail to learn appropriate weights for sampling and learning, rendering them unsuitable for these applications. In addition to these results, statistical tests on the NIST test suite corroborate our findings that sMTJ-clocked LFSRs and high-quality PRNGs such as Xoshiro outperform the pure LFSR-based p-bits (see Supplementary Note XII).

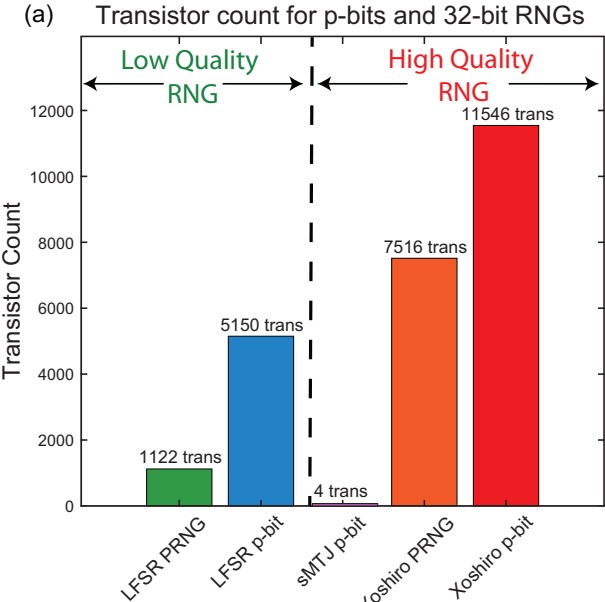

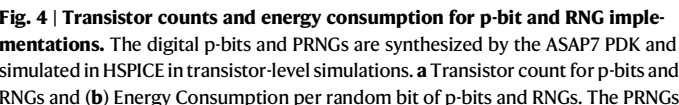

**Fig. 4 | Transistor counts and energy consumption for p-bit and RNG implementations.** The digital p-bits and PRNGs are synthesized by the ASAP7 PDK and simulated in HSPICE in transistor-level simulations. **a** Transistor count for p-bits and RNGs and (**b**) Energy Consumption per random bit of p-bits and RNGs. The PRNGs are 32-bits long and LUTs store $2^8$ words that are 32-bits long to be compared with 32-bit RNGs. The sMTJ-based p-bit result is repeated from ref. 28. To activate the LUT, a periodic input signal with low inputs to the p-bit has been used. See the text and Supplementary Note XV for details on the energy calculation.

Our learning result demonstrates how asynchronously interacting p-bits can creatively combine with existing CMOS technology. Scaled and integrated implementations of this concept could lead to a resurgence in training powerful DBMs[58].

## Energy and transistor count comparisons

Given our prior results stressing how the quality of randomness can play a critical role in probabilistic inference and learning, it is beneficial to perform precise, quantitative comparisons with the various digital PRNGs we built in hardware FPGAs with sMTJ-based p-bits[15]. Note that for this comparison, we do not consider augmented CMOS p-bits, but directly compare sMTJ-based mixed signal p-bits with their digital counterparts (see Supplementary Note XVI for details on projections of integrated implementations using sMTJ-based mixed signal p-bits). Moreover, instead of benchmarking the voltage comparator-based p-bit circuit shown in Fig. 1 or other types of spin-orbit torque-based p-bits[3,59], we benchmark the 3T/1MTJ-based p-bit first reported in ref. 15. The reason for this choice is that this design allows the use of fast in-plane sMTJs whose fluctuations can be as fast as micro to nanoseconds. We also note that the table-top components we use in this work are not optimized but used for convenience.

For the purpose of benchmarking and characterization, we synthesize circuits for LFSR and Xoshiro PRNGs and these PRNG-based p-bits using the ASAP 7nm Predictive process design kit (PDK) that uses SPICE-compatible FinFET device models[60]. Our synthesis flow, explained in detail in Supplementary Note XII, starts from hardware description level (HDL) coding of these PRNGs and leads to transistor-level circuits using the experimentally benchmarked ASAP 7nm PDK. As such, the analysis we perform here offers a high degree of precision in terms of transistor counts and quantitative energy consumption.

Figure 4a shows the transistor count for p-bits using 32-bit PRNGs. Three pieces make up a digital p-bit: PRNG, LUT (for the activation function) and a digital comparator (typically small). To understand how each piece contributes to the transistor count, we separate the PRNG from the LUT contributions in Fig. 4a.

First, we reproduce earlier results reported in ref. 28, corresponding to the benchmarking of the design reported in ref. 15 and

find that a 32-bit LFSR requires 1122 transistors which is very close to the custom-designed 32-bit LFSR with 1194 transistors in ref. 28. However, we find that the addition of an LUT, ignored in ref. 28, adds significantly more transistors. Even though the inputs to the p-bit are 10-bits (s[6][3]), the saturating behavior of the tanh activation allows reductions in LUT size. In our design, the LUT stores $2^8$ words of 32-bit length that are compared to the 32-bit PRNG. Under this precision, the LUT increases the transistor count to 5150, and more would be needed for finer representations. Note that the compact sMTJ-based p-bit design proposed in ref. 15 uses 3 transistors plus an sMTJ which we estimate as having an area of 4 transistors, following ref. 28. In this case, there is no explicit need for a LUT or a PRNG.

Additionally, the results presented in Figs. 2 and 3 indicate that to match the performance of the sMTJ-based p-bits, more sophisticated PRNGs like Xoshiro must be used. In this case, merely the PRNG cost of a 32-bit Xoshiro is 7516 transistors. The LUT costs are the same as LFSR-based p-bits which is about ≈4029 transistors.

Collectively, these results indicate that to truly replicate the performance of an sMTJ-based p-bit, the actual transistor cost of a digital design is ~11,000 transistors which is an order of magnitude worse than the conservative estimation performed in ref. 28.

In Fig. 4b, we show the energy costs of these differences. We focus on the energy required to produce one random bit. Once again, our synthesis flow, followed by ASAP7-based HSPICE simulations, reproduces the results presented in ref. 28. We estimate a 23 fJ energy per random bit from the LFSR-based PRNG where this number was reported to be 20 fJ in ref. 28.

Similar to the transistor count analysis, we consider the effect of the LUT on the energy consumption, which was absent in ref. 28. We first observe that if the LUT is not active, i.e., if the input $I_i$ to the p-bit is not changing, the LUT does not change the energy per random bit very much. In a real p-circuit computation, however, $I_i$ would be continuously changing activating the LUT repeatedly. To simulate these working conditions, we create a variable $I_i$ pulse that wanders around the stochastic window of the p-bit by changing the least significant bits of the input (see Supplementary Note XV). We choose a 1 GHz frequency for this pulse mimicking an sMTJ with a lifetime of 1 ns. We

observe that in this case, the total energy to create a random bit on average increases by a factor of 6× for the LFSR, reaching 145 fJ per bit.

For the practically more relevant Xoshiro, the average consumption per random bit reaches around 293 fJ. Once again, we conclude that the 20 fJ per random bit, reported in ref. 28 underestimates the costs of RNG generation by about an order of magnitude when the RNG quality and other peripheries such as LUTs are carefully taken into account. In this paper, we do not reproduce the energy estimation of the sMTJ-based p-bit but report the estimate in ref. 28, which assumes an sMTJ-based p-bit with ≈ nanosecond fluctuations.

Our benchmarking results highlight the true expense of high-quality digital p-bits in silicon implementations. Given that functionally interesting and sophisticated p-circuits require above 10,000 to 50,000 p-bits[5], using a 32-bit Xoshiro-based p-bit in a digital design would consume up to 0.1 to 0.5 Billion transistors, just for the p-bits. In addition, the limitation of not being able to parallelize or fit more random numbers in hardware would limit the throughput[61] and the probabilistic flips per second, a key metric measuring the effective sampling speed of a probabilistic computer (see for example, refs. 62–64). As discussed in detail in Supplementary Note XVI, near-term projections with $N = 10^4$ p-bits using sMTJs with in-plane magnetic anisotropy (IMA) ($\tau \approx 1\,\text{ns}$[19]) can reach $\approx 10^4$ flips/ns in sampling throughput. These results clearly indicate that a digital solution beyond 10,000 to 50,000 p-bits, as required by large-scale optimization, probabilistic machine learning, and optimization tasks, will remain prohibitive. To solve these traditionally expensive but practically useful problems, the heterogeneous integration of sMTJs holds great promise both in terms of scalability and energy efficiency.

## Discussions

This work demonstrates the first hardware demonstration of a heterogeneous computer combining versatile FPGAs with stochastic MTJs for probabilistic inference and deep Boltzmann learning. We introduce a new variation-tolerant p-bit circuit that is used to create an asynchronous clock domain, driving digital p-bits in the FPGA. In the process, the CMOS + sMTJ computer shows how commonly used and inexpensive PRNGs can be augmented by magnetic nanodevices to perform as well as high-quality PRNGs (without the resource overhead), both in probabilistic inference and learning experiments. Our CMOS + sMTJ computer also shows the first demonstration of training a deep Boltzmann network in a 32-node Chimera topology, leveraging the asynchronous dynamics of sMTJs. Careful comparisons with existing digital circuits show the true potential of integrated sMTJs, which can be scaled up to million p-bit densities far beyond the capabilities of present-day CMOS technology (see Supplementary Note XVI for detailed benchmarking and a p-computing roadmap).

## Methods

### sMTJ fabrication and circuit parameters

We employ a conventional fixed and free layer sMTJ, both having perpendicular magnetic anisotropy. The reference layer thickness is 1 nm (CoFeB) while the free layer is 1.8 nm (CoFeB), deliberately made thicker to reduce its energy barrier[28,35]. The stack structure of the sMTJs we use is, starting from the substrate side, Ta(5)/Pt(5)/[Co(0.4)/Pt(0.4)]$_6$/Co(0.4)/Ru(0.4)/[Co(0.4)/Pt(0.4)]$_2$/Co(0.4)/Ta(0.2)/CoFeB(1)/MgO(1.1)/CoFeB(1.8)/Ta(5)/Ru(5), where the numbers are in nanometers (Fig. 1a). Films are deposited at room temperature by dc/rf magnetron sputtering on a thermally oxidized Si substrate. The devices are fabricated into a circular shape with a 40–80 nm diameter using electron beam lithography and Ar ion milling and annealed at 300 °C for 1 hour by applying a 0.4 T magnetic field in the perpendicular direction. The average tunnel magnetoresistance ratio (TMR) and resistance area product (RA) are 65% and 4.7 Ω μm², respectively. The discrete sMTJs used in this work are first cut out from the wafer, and the electrode pads of the sMTJs are bonded with wires to IC sockets. The following parameters are measured by sweeping DC current to the sMTJ and measuring the voltage. The resistance of the P state $R_P$ is 4.4–5.7 kΩ, the resistance of the AP state $R_{AP}$ is 5.9–7.4 kΩ, and the current at which P/AP fluctuations are 50% is defined as $I_{50/50}$, in between 14–20 μA. At the output of the new p-bit design, we use an extra branch with a bipolar junction transistor that acts as a buffer to the peripheral module pins of the Kintex UltraScale KU040 FPGA board. Given the electrostatic sensitivity of the sMTJs, this branch also protects the circuit from any transients that might originate from the FPGA.

### Digital synthesis flow

HDL codes are converted to gate-level models using the Synopsys Design Compiler. Conversion from these models to Spice netlists is done using Calibre Verilog-to-LVS. Netlist post-processing is done by a custom Mathematica script to make it HSPICE compatible. Details of the synthesis flow (shown in Fig. 4), followed by HSPICE simulation results for functional verification and power analysis are provided in Supplementary Notes XIII, XIV, and XV.

## Data availability

All processed data generated in this study are provided in the main text and Supplementary Information. The data that support the plots within this paper and other findings of this study are available from the corresponding author upon request.

## Code availability

The computer code used in this study is available from the corresponding author upon request.

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

## Acknowledgements

We are grateful to Subhasish Mitra and Carlo Gilardi for discussions regarding LFSRs and high-level synthesis. We gratefully acknowledge Kevin Cao and Mishel Jyothis Paul for their help with the configuration of ASAP7 PDK. We are grateful to Shuvro Chowdhury for his comments on an earlier version of this manuscript. The U.S. National Science Foundation (NSF) grant CCF 2106260, the Office of Naval Research Young Investigator Program (YIP) grant, SAMSUNG Global Research Outreach (GRO) grant, and an NSF CAREER grant are acknowledged by N.S.S., Q.C., K.S., T.H., S.N., N.A.A., and K.Y.C. for supporting this research. Murata Science Foundation and Marubun Research Promotion Foundation are acknowledged by K.K. JST-CREST Grant No. JPMJCR19K3, JST-AdCORP Grant No. JPMJKB2305 and MEXT X-NICS Grant No. JPJ011438 are acknowledged by S.F. JST-PRESTO Grant No. JPMJPR21B2 is acknowledged by S.K.

## Author contributions

K.Y.C. and S.F. conceived and supervised the study. N.S.S. developed the ASAP7 synthesis flow, ran SPICE simulations, and performed circuit-level experiments with sMTJs along with K.K., Q.C., and K.S. K.K., S.K., S.F., and H.O. fabricated sMTJs. K.K., Q.C., and S.K. ran the device-level sMTJ experiments. N.A.A., S.N., and T.H. have implemented the FPGA design for the learning and inference experiments. All authors have discussed the results and participated in writing and improving the manuscript.

## Competing interests

The authors declare no competing Interests.
