## [Peer Review File · Nature Communications]

CMOS + stochastic nanomagnets: heterogeneous computers for probabilistic inference and learningEditorial Note: This manuscript has been previously reviewed at another journal that is not operating a transparent peer review scheme. This document only contains reviewer comments and rebuttal letters for versions considered at *Nature Communications*.

REVIEWER COMMENTS

Reviewer #1 (Remarks to the Author):

This manuscript demonstrates an sMTJ/CMOS hybrid stochastic computer. Compared with high-quality CMOS PRNG, this hybrid computer achieves the same quality of random number generation while using less transistor count and energy. The route of CMOS+sMTJ for stochastic computing is innovative and meaningful, although the current implementation of CMOS+sMTJ is relatively primitive. The authors provide sufficient explanation for our comments and make some improvements in the manuscript. I think it is suitable for publication in *Nature Communications*.

Reviewer #2 (Remarks to the Author):

Summary:

The manuscript presents an innovative approach to extend Moore's Law by integrating stochastic magnetic tunnel junction (sMTJ) with CMOS technology, aiming to enhance Monte Carlo algorithms for applications in machine learning, optimization, and quantum simulation. The authors propose a CMOS + sMTJ prototype that combines sMTJ-based probabilistic bits (p-bits) with Field Programmable Gate Arrays (FPGA), claiming to achieve high-quality true randomness and energy-efficient probabilistic computing.

Major Comments:

a. Novelty and Significance: The concept of augmenting CMOS with emerging nanotechnologies is a promising direction for advancing probabilistic computing. The authors' approach to integrating sMTJ-based p-bits with FPGA is novel and could have a significant impact on the fields mentioned. The claim of replacing 10,000 transistors with compact sMTJ-based p-bits while reducing energy consumption is particularly noteworthy.

b. Comparison and Benchmarking: The comprehensive comparison with LFSR and Xoshiro is appreciated. However, the manuscript would benefit from additional benchmarking against current state-of-the-art probabilistic computing technologies.

Besides, the provided experiments and comparisons regard power consumption and task efficacy, which are commendable for their depth and thoroughness. However, a significant concern arises from the lack of evaluation concerning latency aspects.

The manuscript indicates that the generation of random numbers by the proposed method operates at the millisecond level. While this may be suitable for certain applications, the adequacy of such speeds for large-scale deployment remains unsubstantiated. The practicality of this method in real-world scenarios where rapid generation of random numbers is crucial needs to be convincingly demonstrated.

To address this gap, I strongly recommend that the authors include an evaluation of latency performance, particularly in the context of typical large-scale applications that rely on random number generation. This could be achieved through simulation-based studies or analytical modeling that provide insights into the expected delays when scaling up the

technology. Such an analysis would greatly enhance the manuscript's contribution by clarifying the potential limitations or confirming the scalability of the proposed solution.

Minor Comments:

- a. The authors should consider discussing the scalability of their approach in more detail, including potential challenges and solutions.
- b. A discussion on the integration of the proposed technology with existing computing infrastructure would be valuable.
- c. the manuscript's conclusion is generally positive, but it does not delve into any specifics about potential limitations or future directions that could be taken to further develop or apply the proposed approach. It would be useful if the authors could provide some additional insights or suggestions about future research directions in this area.

Conclusion:

The manuscript offers a potentially groundbreaking approach to probabilistic computing. While the manuscript excels in certain experimental areas, it falls short in providing a holistic view of the technology's performance. Additional experimental details, further validation, and comprehensive benchmarking are necessary to substantiate the claims made. A revision that incorporates latency evaluations is necessary to ensure a comprehensive understanding of the method's applicability in large-scale settings.

In conclusion, I recommend a revision with the above points addressed before considering publication in Nature Communications.

Reviewer's Recommendation: Major Revision

Reviewer #3 (Remarks to the Author):

Revisions to Kobayashi et al.

Considering the current revision of submission now from the perspective of a different journal; I continue to see the value of this study and can see that it is near suitable for acceptance, yet I still remain concerned about the messaging of the paper being clear. In the response to reviewers, the authors make a strong point about how their paper is about more than just improving on PRNGs. However, upon reading the paper again, that still seems to be the primary message – the coupling of sMTJ p-bits into an FPGA circuit can be used to do effective random sampling. That they can use even low-quality PRNGs to boost their impact is important as well. These are good results and worth publishing once the story is crisp.

While I am appreciative of the other notes that the authors list in their first response – that this is a step towards asynchronous p-bit circuits, that they are introduce a new type of p-bit centric Boltzmann machine, and their novel p-bit circuit – I would encourage the authors to ask themselves why these messages got lost in their main text of the document. They're all nice aspects of the work, but they're all also rather specific claims that require a much deeper investigation to stand on their own; their value seems more to support the overall narrative – that hybrid CMOS+sMTJ circuits can be used efficiently for sampling.

All this is to say that the paper, as is, is near sufficient to publish. But I would encourage the authors to look closely at the paper themselves and ask if it is the most effective at communicating the message that they want to convey. In fact, now that this is the third(?) time I have looked at this paper, I wonder if the story is being told backwards.

One suggestion would be to look to their conclusion to help frame the introduction. The authors make a strong point in the penultimate paragraph of their paper, when they state

that “These results clearly indicate that a digital solution beyond 10000 to 50000 p-bits will remain prohibitive and the heterogeneous integration of sMTJs holds great promise both in terms of scalability and energy-efficiency.”

Some sentence to this effect needs to be very clear in the introduction. The average reader of this in Nature Communications, while perhaps familiar with the concept of p-bits, is not going to be appreciative of the fact that (1) real world applications of p-bits likely will require an extremely large number of p-bits to be instantiated in parallel, (2) the cost of doing this conventionally more or less will defeat the point, and (3) fully hardware-based solutions for stochasticity at that scale are a long way off. I believe the authors can state this as motivation without tipping their hand on the results in Figure 4. As a result, the design of the intermediate hybrid solution that they explore here is more than just an illustration of p-bits (which is kind of repeating past work) or an evaluation of the quality of conventional PRNGs (which is where my reviews have been focused). Rather, the story is meant to be arguing that this is an important step towards making p-bits a viable technology. I believe that all of the results – the robust hybrid circuit, the deep Boltzmann learning, the asynchrony, the estimation of costs – all clearly fall into line on this vision. I'd further encourage them to set up those stories in the introduction as well as necessary steps towards advancing the technology.

As it is currently written, the introduction reads like a fairly generic introduction to the idea of a hybrid architecture and saying p-bits are exciting. I'd strongly suggest that the authors step back and ensure that their introduction (and abstract) are effective at highlighting not only what they are trying to do, but why. Because in this current submission the “why” is not obvious.

CMOS + stochastic nanomagnets: heterogeneous computers for probabilistic inference and learning

Nihal Sanjay Singh,^{*} Keito Kobayashi,^{*} Qixuan Cao,^{*} Kemal Selcuk, Tianrui Hu, Shaila Niazi,
Navid Anjum Aadit, Shun Kanai, Hideo Ohno, Shunsuke Fukami,[†] and Kerem Y. Camsari[‡]

I. REVIEWER 2

- **#2.1 Summary:** *The manuscript presents an innovative approach to extend Moore’s Law by integrating stochastic magnetic tunnel junction (sMTJ) with CMOS technology, aiming to enhance Monte Carlo algorithms for applications in machine learning, optimization, and quantum simulation. The authors propose a CMOS + sMTJ prototype that combines sMTJ-based probabilistic bits (p-bits) with Field Programmable Gate Arrays (FPGA), claiming to achieve high-quality true randomness and energy-efficient probabilistic computing. Major Comments: a. Novelty and Significance: The concept of augmenting CMOS with emerging nanotechnologies is a promising direction for advancing probabilistic computing. The authors’ approach to integrating sMTJ-based p-bits with FPGA is novel and could have a significant impact on the fields mentioned. The claim of replacing 10,000 transistors with compact sMTJ-based p-bits while reducing energy consumption is particularly noteworthy.*

AUTHOR RESPONSE

We appreciate the accurate summary. We agree with the reviewer’s key observation that the present work is an important milestone to advance probabilistic computing.

-
- **#2.2 b. Comparison and Benchmarking:** *The comprehensive comparison with LFSR and Xoshiro is appreciated. However, the manuscript would benefit from additional benchmarking against current state-of-the-art probabilistic computing technologies. Besides, the provided experiments and comparisons regard power consumption and task efficacy, which are commendable for their depth and thoroughness. However, a significant concern arises from the lack of evaluation concerning latency aspects. The manuscript indicates that the generation of random numbers by the proposed method operates at the millisecond level. While this may be suitable for certain applications, the adequacy of such speeds for large-scale deployment remains unsubstantiated. The practicality of this method in real-world scenarios where rapid generation of random numbers is crucial needs to be convincingly demonstrated. To address this gap, I strongly recommend that the authors include an evaluation of latency performance, particularly in the context of typical large-scale applications that rely on random number generation. This could be achieved through simulation-based studies or analytical modeling that provide insights into the expected delays when scaling up the technology. Such an analysis would greatly enhance the manuscript’s contribution by clarifying the potential limitations or confirming the scalability of the proposed solution.*

AUTHOR RESPONSE

We appreciate the reviewer’s positive comments regarding power consumption. We do agree with the need for a thorough benchmarking on latency (or delay) regarding p-computers to substantiate the practicality of our approach for large-scale deployment. This is an emerging and fascinating area with many new results. **In a remastered Supplementary Table S4, we provide three levels of latency (delay) analysis:**

^{*} These authors contributed equally

[†] shunsuke.fukami.c8@tohoku.ac.jp

[‡] camsari@ece.ucsb.edu

- **Devices:** Delays at level of the sMTJs
- **Circuits:** Delays at the level of p-circuits made out of sMTJs
- **Systems:** Delays at the level of networks made out of p-circuits.

In each case, we discuss the experimental state of the art and our assumptions behind projections to substantiate our claims. Table S4 (reproduced for convenience below) presents a comparative benchmarking at these three levels. To summarize our findings and report:

- **Device level:** Recent experimental breakthroughs have shown that nanosecond fluctuations (up to GHz speeds) can be obtained via sMTJs with *in-plane* magnetic anisotropy [1–3]. These results are a significant departure from earlier work on sMTJs with *perpendicular* magnetic anisotropy, establishing that p-bits with billions of flips per second are feasible using present-day technology.
- **Circuit level:** Even though tunable binary stochastic neurons, in this context, p-bits, can be constructed using fast sMTJs. These bring in additional experimental challenges. First, the electronic circuitry must be faster than the fluctuation rates of sMTJs to be able to resolve each flip. In principle, CMOS transistors can work with picosecond time scales, as such, GHz fluctuations of sMTJs should not pose a fundamental problem. In practice, however, the fastest p-bits demonstrated to date (reported recently in [4]) work with \approx microsecond time scales. Even though these are about 3 orders of magnitude faster than earlier \approx millisecond p-bits, they show how the path to fast p-bits is a rapidly advancing field.
- **System level:** At the system level, where we imagine large-scale sMTJ-based p-bits will be integrated with CMOS, the key metric for p-computers will be the amount of probabilistic decisions (flips) per second. For a given application, in sampling or optimization, a large amount of serially conditioned flips need to be taken. In our earlier work, we showed that *asynchronous* [5] or pseudo-asynchronous [6] architectures can retain ideal massive parallelism where the total number of flips taken by a network scales proportionally with the number of p-bits in the network.

For this to materialize however, two important requirements need to hold:

1. Requirement #1: The number of p-bits integrated on chip must be large, $N \approx 10^5$ to 10^6 to reach high-throughput. Considering present-day magnetic RAM technology, integrated MTJs up to billion bit densities have been fabricated.
2. Requirement #2: The “synapse” time calculation needs to be faster than the fastest p-bit in the system. Considering native CMOS speeds or resistive crossbars made to work with stochastic MTJs [7] with about picosecond latencies, nanosecond fluctuations of MTJs should be well-above the synapse time requirements.

AUTHOR ACTION

- Please see a revised Section XVI in the Supplementary Information where we summarize the above information of the study of latencies from device, circuit and systems perspectives.

TABLE S4. Benchmarking probabilistic hardware from device, circuit and system perspectives. For device comparisons we focus on experimentally demonstrated sMTJ fluctuations. p-bits are circuits that use sMTJs to produce binary stochastic neurons with tunable probability with fluctuations at τ^{-1} rates. At the system level, we focus on number of p-bits in a network (N) and sampling throughput, which is given by N/τ , for asynchronous systems with fast synapses computing Eq. S.2 (see text). Also at the system level, we report published data for GPUs/TPUs handling similar probabilistic sampling tasks. Projections are shown using †. (*) In heterogeneous computers of the type we consider in this paper, external sMTJ-based p-bits can drive a large number of digital p-bits in an FPGA or an ASIC. ° RNG quality is deemed low / high for *sampling* problems rather than combinatorial optimization problems for which LFSR-based PRNGs seem sufficient [6].

Level of Comparison	GPUs/TPUs	← THIS WORK →				NEAR-TERM PROJECTION		LONG-TERM PROJECTION
		FPGA p-computer		Hetero FPGA+sMTJ p-computer		ASIC all-digital	Heterogeneous sMTJ+ASIC	
		LFSR-based	Xoshiro-based	PMA	IMA	IMA		
DEVICE: sMTJ	N/A	N/A	N/A	$\tau \approx 1 \text{ ms}$ [8]	$\tau \approx 1 \mu\text{s}$ [9]	$\tau \approx 1 \text{ ns}$ [1]	N/A	$\tau \approx 1 \text{ ns}^\dagger$
CIRCUIT: p-bit	N/A	$(\tau)^{-1} \approx 10 \text{ MHz}$	$(\tau)^{-1} \approx 10 \text{ MHz}$	$\tau \approx 1 \text{ ms}$ [8]	$\tau \approx 1 \mu\text{s}$ [4]	$\tau \approx 1 \text{ ns}^\dagger$	$\tau \approx 1 \text{ ns}^\dagger$	$\tau \approx 1 \text{ ns}^\dagger$
SYSTEM: Number of p-bits (N)	N/A	$N \approx 10^4$ [6]	$N \approx 10^4$ [10]	$N = 32(*)$	$N \approx 10^4(*)^\dagger$	$N \approx 10^4(*)^\dagger$	$N \approx 10^5^\dagger$	$N \approx 10^6^\dagger$
SYSTEM: Sampling throughput (flips/ns)	≈ 10	≈ 100	≈ 100	$\approx 32 \times 10^{-6}$	$\approx 10^\dagger$	$\approx 10^4^\dagger$	$\approx 10^5^\dagger$	$\approx 10^6^\dagger$
RNG Quality	PRNG/ High	PRNG/ Low°	PRNG/ High	TRNG/ High	TRNG/ High	TRNG/ High	PRNG/ High	PRNG/ High

-
- **#2.3 Minor Comments:** a. The authors should consider discussing the scalability of their approach in more detail, including potential challenges and solutions.

AUTHOR RESPONSE

We appreciate this suggestion. Scalability of our approach from a device, circuit, and systems perspective has been revised in an updated introduction and Section XVI, in the supplementary information.

AUTHOR ACTION

- Please see the revised introduction, discussing these aspects at a high level.
- Please see the revised Supplementary Section XVI for a more detailed look into the scalability-related challenges and solutions.

-
- **#2.4 b.** A discussion on the integration of the proposed technology with existing computing infrastructure would be valuable.

AUTHOR RESPONSE/ACTION

Please see the revised Table S4 and the associated text where we discuss integration with existing CMOS infrastructure.

-
- **#2.5 c.** the manuscript's conclusion is generally positive, but it does not delve into any specifics about potential limitations or future directions that could be taken to further develop or apply the proposed approach. It would be useful if the authors could provide some additional insights or suggestions about future research directions in this area.

AUTHOR RESPONSE/ACTION

We have revised our introduction and the conclusion section where we discuss future research directions.

- **#2.6 Conclusion:** *The manuscript offers a potentially groundbreaking approach to probabilistic computing. While the manuscript excels in certain experimental areas, it falls short in providing a holistic view of the technology's performance. Additional experimental details, further validation, and comprehensive benchmarking are necessary to substantiate the claims made. A revision that incorporates latency evaluations is necessary to ensure a comprehensive understanding of the method's applicability in large-scale settings. In conclusion, I recommend a revision with the above points addressed before considering publication in Nature Communications.*

AUTHOR RESPONSE

We appreciate the reviewer's summary of their concerns. Please see our responses to their specific concerns above. We believe that our revised manuscript discusses all relevant experimental details, validation and benchmarking with existing technology and what scaled p-computers may entail in large scale settings.

We hope that the changes made, after several rounds of revision by the reviewer with a multitude of different aspects being examined are now satisfactory.

II. REVIEWER 3

- **#3.0** *Considering the current revision of submission now from the perspective of a different journal; I continue to see the value of this study and can see that it is near suitable for acceptance, yet I still remain concerned about the messaging of the paper being clear. In the response to reviewers, the authors make a strong point about how their paper is about more than just improving on PRNGs. However, upon reading the paper again, that still seems to be the primary message – the coupling of sMTJ p-bits into an FPGA circuit can be used to do effective random sampling. That they can use even low-quality PRNGs to boost their impact is important as well. These are good results and worth publishing once the story is crisp.*

AUTHOR RESPONSE

We thank the reviewer for the thoughtful review. We are happy to see we have made progress in the previous rounds.

- **#3.1** *While I am appreciative of the other notes that the authors list in their first response – that this is a step towards asynchronous p-bit circuits, that they are introduce a new type of p-bit centric Boltzmann machine, and their novel p-bit circuit – I would encourage the authors to ask themselves why these messages got lost in their main text of the document. They're all nice aspects of the work, but they're all also rather specific claims that require a much deeper investigation to stand on their own; their value seems more to support the overall narrative – that hybrid CMOS+sMTJ circuits can be used efficiently for sampling.*

One suggestion would be to look to their conclusion to help frame the introduction. The authors make a strong point in the penultimate paragraph of their paper, when they state that “These results clearly indicate that a digital solution beyond 10000 to 50000 p-bits will remain prohibitive and the heterogeneous integration of sMTJs holds great promise both in terms of scalability and energy-efficiency.”

Some sentence to this effect needs to be very clear in the introduction. The average reader of this in Nature Communications, while perhaps familiar with the concept of p-bits, is not going to be appreciative of the fact that (1) real world applications of p-bits likely will require an extremely large number of p-bits to be instantiated in parallel, (2) the cost of doing this conventionally more or less will defeat the point, and (3) fully hardware-based solutions for stochasticity at that scale are a long way off. I believe the authors can state this as motivation without tipping their hand on the results in Figure 4. As a result, the design of the intermediate hybrid solution that they explore here is more than just an illustration of p-bits (which is kind of repeating past work) or an evaluation of the quality of conventional PRNGs (which is where my reviews have been focused). Rather, the story is meant to be arguing that this is an important step towards making p-bits a viable technology. I believe that all of the results – the robust hybrid circuit, the deep Boltzmann learning, the asynchrony, the estimation of costs – all clearly fall into line on this vision. I'd further encourage them to set up those stories in the introduction as well as necessary steps towards advancing the technology.

As it is currently written, the introduction reads like a fairly generic introduction to the idea of a hybrid architecture and saying p-bits are exciting. I'd strongly suggest that the authors step back and ensure that their introduction (and abstract) are effective at highlighting not only what they are trying to do, but why. Because in this current submission the “why” is not obvious.

AUTHOR RESPONSE

We appreciate these thoughtful comments. We believe that the reviewer is making ONE main point in their response as such, we respond to it as a whole. We also strongly agree with the reviewer's concern that as written, our introduction did not do justice to the main message we are trying to convey. **We have tried to remedy this in a completely re-written (and expanded) introduction.** Below, we present excerpts from our new introduction focused on addressing the concerns laid out by the reviewer.

Despite promising progress with hardware prototypes [7, 8, 11–13], large-scale probabilistic computing using stochastic nanodevices remains elusive. As we will establish in this paper, designing purely CMOS-based high-performance probabilistic computers suited to sampling and optimization problems is prohibitive beyond a certain scale ($>1M$ p-bits) due to the large area and energy costs of pseudorandom number generators. As such, in our view, any large scale integration of probabilistic computing will involve strong integration with CMOS technology, in CMOS+X architectures.

Given the unavoidable device-to-device variability, the interplay between continuously fluctuating stochastic nanodevices (e.g., sMTJs) with deterministic CMOS circuits and possible applications of such hybrid circuits remain unclear.

In this paper, we first introduce the notion of a hybrid CMOS+sMTJ system where the asynchronous dynamics of sMTJs control digital circuits in a standard CMOS Field Programmable Gate Array (FPGA). We view the FPGA as a “drop-in replacement” for eventual integrated circuits where sMTJs could be situated on top of CMOS. Unlike earlier implementations where sMTJs were primarily used to implement neurons and CMOS or analog components circuits for synapses [8, 11], we design hybrid circuits where sMTJ-based p-bits control a large number of digital circuits residing in the FPGA without dividing the system into neurons (sMTJ) and synapses (CMOS). We show how the true randomness injected into deterministic CMOS circuits augment low-quality random number generators based on linear feedback shift registers (LFSR). This result represents an example of how sMTJs could be used to reduce footprint in the CMOS underlayer. Even though we present a rather small example of a CMOS + sMTJ system, we believe that our findings could lay the groundwork for larger implementations in the presence of unavoidable device-to-device variations. We also focus beyond the common use case of combinatorial optimization of similar physical computers [14], considering probabilistic inference and learning in deep energy-based models.

Specifically, we use our system to train 3-hidden 1-visible layer deep and unrestricted Boltzmann machines that entirely rely on the asynchronous dynamics of the stochastic MTJs. Second, we evaluate the quality of randomness directly at the application level through probabilistic inference and deep Boltzmann learning. This approach contrasts with the majority of related work, which typically conducts statistical tests at the single device level to evaluate the quality of randomness [3, 15–19]. As an important new result, we find that the quality of randomness matters in machine learning tasks as opposed to optimization tasks that have been explored previously. And finally, we conduct a comprehensive benchmark using an experimentally calibrated 7-nm CMOS PDK and find that when the quality of randomness is accounted for, the sMTJ-based p-bits are about 4 orders of magnitude smaller in area and they dissipate 2 orders of magnitude less energy, compared to CMOS p-bits. We envision that large-scale purely CMOS-based p-computers ($\gg 10^5$) can be a reality in scaled up versions of the CMOS + sMTJ type computers we discuss in this work.

As the reviewer carefully articulates, our goal has been to emphasize the following two main points:

1. For any successful scaled implementation of probabilistic circuits, hybrid architectures where probabilistic hardware is integrated with CMOS devices will be of crucial importance. In this paper, we have shown how a class of asynchronous computers driven entirely by the stochastic dynamics of sMTJs can be used to sample from traditionally hard-to-sample distributions, which ties to powerful generative machine learning algorithms like deep Boltzmann machines. Beyond comparisons to PRNGs and quality of random numbers, we sought to establish how a digital circuit entirely controlled by sMTJs with varying fluctuation rates (including experimental variation) can learn a deep Boltzmann machine.
2. A secondary, albeit noteworthy point, is to see how otherwise insufficient PRNGs can be “augmented” by sMTJs. This may have implications on just how much CMOS is needed in eventual CMOS + sMTJ architectures.

We believe that the manuscript, as currently revised makes these two points very clear throughout every section.

AUTHOR ACTION

- Please see the revised introduction that details the main points of this work.
 - Please see the revised Supplementary Section XVI for a more detailed look into the scalability-related challenges and solutions.
-

REFERENCES

- [1] Christopher Safranski, Jan Kaiser, Philip Trouilloud, Pouya Hashemi, Guohan Hu, and Jonathan Z Sun. Demonstration of nanosecond operation in stochastic magnetic tunnel junctions. *Nano Letters*, 21(5):2040–2045, 2021.
- [2] Keisuke Hayakawa, Shun Kanai, Takuya Funatsu, Junta Igarashi, Butsurin Jinnai, WA Borders, H Ohno, and S Fukami. Nanosecond random telegraph noise in in-plane magnetic tunnel junctions. *Physical Review Letters*, 126(11):117202, 2021.
- [3] Leo Schnitzspan, Mathias Kläui, and Gerhard Jakob. Nanosecond true-random-number generation with superparamagnetic tunnel junctions: Identification of joule heating and spin-transfer-torque effects. *Phys. Rev. Appl.*, 20:024002, Aug 2023.
- [4] Nihal Sanjay Singh, Shaila Niazi, Shuvro Chowdhury, Kemal Selcuk, Haruna Kaneko, Keito Kobayashi, Shun Kanai, Hideo Ohno, Shunsuke Fukami, and Kerem Yunus Camsari. Hardware demonstration of feedforward stochastic neural networks with fast mtj-based p-bits. In *2023 International Electron Devices Meeting (IEDM) Proceedings*. IEEE, 2023.
- [5] Navid Anjum Aadit, Andrea Grimaldi, Giovanni Finocchio, and Kerem Y Camsari. Physics-inspired ising computing with ring oscillator activated p-bits. In *2022 IEEE 22nd International Conference on Nanotechnology (NANO)*, pages 393–396. IEEE, 2022.
- [6] Navid Anjum Aadit, Andrea Grimaldi, Mario Carpentieri, Luke Theogarajan, John M Martinis, Giovanni Finocchio, and Kerem Y Camsari. Massively parallel probabilistic computing with sparse ising machines. *Nature Electronics*, 5(7):460–468, 2022.
- [7] Sidra Gibeault, Temitayo N Adeyeye, Liam A Pocher, Daniel P Lathrop, Matthew W Daniels, Mark D Stiles, Jabez J McClelland, William A Borders, Jason T Ryan, Philippe Talatchian, et al. Programmable electrical coupling between stochastic magnetic tunnel junctions. *arXiv preprint arXiv:2312.13171*, 2023.
- [8] William A Borders et al. Integer factorization using stochastic magnetic tunnel junctions. *Nature*, 573:390–393, 2019.
- [9] Keito Kobayashi, Keisuke Hayakawa, Junta Igarashi, William A Borders, Shun Kanai, Hideo Ohno, and Shunsuke Fukami. External-field-robust stochastic magnetic tunnel junctions using a free layer with synthetic antiferromagnetic coupling. *Physical Review Applied*, 18(5):054085, 2022.
- [10] Shaila Niazi, Navid Anjum Aadit, Masoud Mohseni, Shuvro Chowdhury, Yao Qin, and Kerem Y Camsari. Training deep boltzmann networks with sparse ising machines. *arXiv preprint arXiv:2303.10728*, 2023.
- [11] Jan Kaiser, William A Borders, Kerem Y Camsari, Shunsuke Fukami, Hideo Ohno, and Supriyo Datta. Hardware-aware in situ learning based on stochastic magnetic tunnel junctions. *Physical Review Applied*, 17(1):014016, 2022.
- [12] Jia Si, Shuhan Yang, Yunuo Cen, Jiaer Chen, Zhaoyang Yao, Dong-Jun Kim, Kaiming Cai, Jerald Yoo, Xuanyao Fong, and Hyunsoo Yang. Energy-efficient superparamagnetic ising machine and its application to traveling salesman problems. *arXiv preprint arXiv:2306.11572*, 2023.
- [13] John Daniel, Zheng Sun, Xuejian Zhang, Yuanqiu Tan, Neil Dilley, Zhihong Chen, and Joerg Appenzeller. Experimental demonstration of an integrated on-chip p-bit core utilizing stochastic magnetic tunnel junctions and 2d-mos $\text{-}\{2\}$ fets. *arXiv preprint arXiv:2308.10989*, 2023.
- [14] Naeimeh Mohseni, Peter L McMahon, and Tim Byrnes. Ising machines as hardware solvers of combinatorial optimization problems. *Nature Reviews Physics*, 4(6):363–379, 2022.
- [15] Damir Vodonicarevic, Nicolas Locatelli, Alice Mizrahi, Joseph S Friedman, Adrien F Vincent, Miguel Romera, Akio Fukushima, Kay Yakushiji, Hitoshi Kubota, Shinji Yuasa, et al. Low-energy truly random number generation with superparamagnetic tunnel junctions for unconventional computing. *Physical Review Applied*, 8(5):054045, 2017.
- [16] Bradley Parks, Mukund Bapna, Julianne Igbokwe, Hamid Almasi, Weigang Wang, and Sara A Majetich. Superparamagnetic perpendicular magnetic tunnel junctions for true random number generators. *AIP Advances*, 8(5):055903, 2018.
- [17] Vaibhav Ostwal and Joerg Appenzeller. Spin-orbit torque-controlled magnetic tunnel junction with low thermal stability for tunable random number generation. *IEEE Magnetics Letters*, 10(4503305):1–5, 2019.
- [18] Yang Lv, Brandon R Zink, and Jian-Ping Wang. Bipolar random spike and bipolar random number generation by two magnetic tunnel junctions. *IEEE Transactions on Electron Devices*, 69(3):1582–1587, 2022.
- [19] Zhenxiao Fu, Yi Tang, Xi Zhao, Kai Lu, Yemin Dong, Amit Shukla, Zhifeng Zhu, and Yumeng Yang. An overview of spintronic true random number generator. *Frontiers in Physics*, 9:638207, 2021.

REVIEWERS' COMMENTS

Reviewer #2 (Remarks to the Author):

The revision has made significant progress in addressing my earlier concerns, resulting in a manuscript with a more comprehensive structure and a detailed evaluation of latency. However, to fully grasp the significance of the proposed "CMOS+sMTJ" method, it would be beneficial to include a clearer illustration of how practical probabilistic applications are constrained by the p-bit scale. This addition would enhance readers' understanding of the method's benefits.

Reviewer #3 (Remarks to the Author):

The authors have taken my comments regarding the framing of the paper into account, especially in the revised introduction, and I am satisfied with the paper in its current form.

Re: NCOMMS-23-46937A

CMOS + stochastic nanomagnets: heterogeneous computers for probabilistic inference and learning

Nihal Sanjay Singh,^{*} Keito Kobayashi,^{*} Qixuan Cao,^{*} Kemal Selcuk, Tianrui Hu, Shaila Niazi,
Navid Anjum Aadit, Shun Kanai, Hideo Ohno, Shunsuke Fukami,[†] and Kerem Y. Camsari[‡]

I. REVIEWER 2

- **#2.1** *The revision has made significant progress in addressing my earlier concerns, resulting in a manuscript with a more comprehensive structure and a detailed evaluation of latency. However, to fully grasp the significance of the proposed "CMOS+sMTJ" method, it would be beneficial to include a clearer illustration of how practical probabilistic applications are constrained by the p-bit scale. This addition would enhance readers' understanding of the method's benefits.*

AUTHOR RESPONSE

We agree. We have clearly stated that our present demonstration is a small-scale prototype and scaled implementations will be needed for bigger and more practical benefits. We have also mentioned the category of problems targeted by these large-scale solutions.

AUTHOR ACTION

- Please see the modified Introduction Section
 - Please see the modified Energy and transistor count comparisons Section
-

^{*} These authors contributed equally

[†] shunsuke.fukami.c8@tohoku.ac.jp

[‡] camsari@ece.ucsb.edu